# Emergent Proteins-Based Structures—Prospects towards Sustainable Nutrition and Functionality

**DOI:** 10.3390/gels7040161

**Published:** 2021-10-01

**Authors:** Ricardo N. Pereira, Rui M. Rodrigues

**Affiliations:** CEB—Centre of Biological Engineering, University of Minho, 4710-057 Braga, Portugal; ruirodrigues@ceb.uminho.pt

**Keywords:** algae, whey, fermentation functionality, emulsion, gels

## Abstract

The increased pressure over soils imposed by the need for agricultural expansion and food production requires development of sustainable and smart strategies for the efficient use of resources and food nutrients. In accordance with worldwide transformative polices, it is crucial to design sustainable systems for food production aimed at reducing environmental impact, contributing to biodiversity preservation, and leveraging a bioeconomy that supports circular byproduct management. Research on the use of emergent protein sources to develop value-added foods and biomaterials is in its infancy. This review intends to summarize recent research dealing with technological functionality of underused protein fractions, recovered from microbial biomass and food waste sources, addressing their potential applications but also bottlenecks. Protein-based materials from dairy byproducts and microalgae biomass gather promising prospects of use related to their techno-functional properties. However, a balance between yield and functionality is needed to turn this approach profitable on an industrial scale basis. In this context, downstream processing should be strategically used and properly integrated. Food solutions based on microbial proteins will expand in forthcoming years, bringing the opportunity to finetune development of novel protein-based biomaterials.

## 1. Introduction

Proteins are among the most satiating food nutrients, providing the needed physiological support of the human body but also enclosing a complex set of dynamic and structural properties that make them transversal to many fields of science, such as food biotechnology, biomedical, pharmaceutical, and cosmetics. In the context of COVID-19 pandemic, the role of food nutrients in the improvement of the immune system is also attracting attention [1,2].

Food proteins, aside from their nutritional importance (by supplying essential amino acids), are used in many food formulations as structuring agents given their gelling, thickening, emulsifying, and foaming properties. Furthermore, many of the most important food proteins (e.g., milk and egg) can be used as “building blocks” for the innovative fabrication of food grade superstructures [3]. These superstructures can be produced at nano- and microscale and used as delivery systems for drugs and bioactive compounds, thus bringing nutritional advantages and targeted health functions. Other emergent area of application is closely related with tissue engineering applications through development of 3D architectures that can support and leverage cellular growth [4]. Paradoxically, diverse human neurodegenerative diseases, such as Alzheimer’s, Parkinson’s, and transmissible spongiform encephalopathies are intrinsically related with changes in the protein folding that lead to misfolded fibrillar structures, known as the amyloid fibrils [5]. These amyloid structures can be designed in vitro as novel biomaterials, using food proteins from animal and plant sources [6], and could potentially be used as low-cost models for testing therapeutic efficacy of drugs and nutraceuticals against amyloidal diseases. In addition to acting as amyloid models, these structures, and the understanding about how to control molecular interactions between food proteins during amyloid assembly and between them and bioactive molecules could bring new dietary strategies with benefits in disease prevention.

Most food proteins are animal-based, and meat consumption is still deep-rooted worldwide and culturally bounded in many societies. Nonetheless, during last decade, meat consumption patterns changed mainly in developed countries. Consumers are now more prone to choose meat-free diets much for health reasons and social consciousness regarding environment and animal welfare issues [7,8]. Nonconventional sources of proteins are now booming in western societies and attracting attention in food research. Proteins from plants, insects, and microorganisms (such as microalgae, yeasts and fungi, bacteria) are expanding in several food applications, and this transition to nonconventional proteins is being harnessed for environmental, economic, political, and social reasons. They offer several advantages, and production is aligned with agricultural and environmental sustainability goals. Microbial biomass is a potential source of functional molecules and can be sustainably produced through fermentation processes using underrated organic feedstock (e.g., byproducts from agri-food industries) for growing single microorganisms and microorganisms’ consortia. Microorganisms have a rapid growth rate, and their cultivation is performed under optimization of biological reactors, thus not requiring the use of arable land, contrary to what happens for crop cultivation and animal husbandry, for example. Microbial and plant-based protein also offer versatility, low environmental footprint, and competitive production However, safety aspects related with potential toxicity or allergenicity still need to be underpinned [9]. Another emergent approach is the valorization of protein-rich byproducts from food industry, integrated by the concept of biorefinery, where several fractions of a given product can be reutilized and meet the emergent challenges of an intended bioeconomy [10]. An outstanding example of this strategy is related to the use of liquid whey, resultant from cheese and tofu production, which instead of being discarded as waste can be used as a source for the development of protein-based functional systems. Outstanding reviews can be found in literature that highlight the potential use of plant-based food proteins as ingredients for food formulation highlighting aspects related with nutritional aspects [11,12]. However, less attention is devoted to development of wise protein systems for food and biomedical applications, as well as to the need of downstream processing technologies to recover and tune protein properties for a tailored function. 

This review highlights recent research about the use of sustainable and inexpensive sources of food proteins for functional food arena, and for the development innovative biomaterials to be used in biomedical applications.

## 2. Promising Protein Sources and their Challenges

The need to reduce the food production environmental footprint is fostering the interest in protein sources that until recently was underestimated. Most promising protein food sources include food waste byproducts (such as whey), algae (micro and macro), and microorganisms such as fungi, yeast and bacteria. Figure 1 illustrates some of these emergent protein sources, highlighting examples of their main characteristics. 

### 2.1. Food Waste

The concept of recovering protein rich fractions from industrial food waste is not novel, but it is gaining increasing interest as way to pursue a sustainable food system [13]. Recent literature points out some of the most promising food waste streams that can offer significant amount of crude protein with a balanced composition regarding their essential amino acid (EEA) profile [13,14]. Plant-based food waste encompasses interesting streams such malted barley germs, brewing cake, papaya seed, and pumpkin kernel cake, which can present amounts of crude protein up to 40%, with 30–40% of EAA in the total amino acid composition [13,14]. In general, byproducts of oil processing, such as press cakes from rapeseed sunflower or canola, may contain 45–65% protein [12]. But dairy waste (from hard and soft cheese yogurt and milk whey) still has one of the best functional properties with considerable amounts of crude protein content (up to 40%) and higher percentage of the nine EAA (up to 45%).

The major bottleneck of these emergent sources concerns the right balance between amount of protein and its nutritional and functional quality without overlooking safety—indeed, some fractions present interesting amounts of protein, but their nutritional value is rather low and vice versa. Most of these streams result from harsh processing involving high temperatures, drying, pH-shift, and fermentation, which may induce different levels of protein denaturation, proteolysis, or even racemization of the EAA. The recovery of proteins from these streams involves the use of extraction strategies, which in turn poses a set of challenges: (i) increase profitability of extraction methodologies at an industrial scale; (ii) reduce the impact of extraction on nutritional and functional aspects of a given protein fraction; (iii) harness a holistic way thinking to leverage functionality, safety, and affordability of these fractions [13,15].

### 2.2. Microbial Protein—Yeast, Fungi, and Bacteria

Microbial protein is gaining in popularity due to its low-cost production, sustainable character, and nutritional profile. In western societies, protein from yeast and fungi are likely to be generally accepted once they already find niche commercial channels and are anciently present in some conventional food products—i.e., beer, cheese yogurt, bread, among others. Several species of fungi and yeast are perceived as superior by the consumer and receive greater acceptance in the food industry, as is the case of *Saccharomyces cerevisiae*, *Penicillium roqueforti*, *Penicillium camemberti*, which are mainly used in the bakery, brewing, and dairy industries. Microorganism consortia can convert industrial food wastes (e.g., sugarcane bagasse, whey, brewery’s spent grains) through liquid or solid state fermentation through a biorefinery approach, allowing to produce protein-enriched biomass with the potential for functional foods [16,17,18]. *Kefir* is an example a product resulting from fermentation of dairy and nondairy substrates (e.g., fruits and molasses) using mixture of bacteria and yeast that brings potential health effects due to obtained myriad of macro- and micronutrients including proteins, as well as pre- and probiotics [19,20]. 

One case of success is mycoprotein produced from the filamentous fungus *Fusarium venenatum*, currently used as meat substitute food ingredient from Quorn trademark. Mycoprotein (known also as fungal protein) is being discussed as nutritious protein source resulting from continuous fermentation of sugars and its high protein and fiber contents (by dry weight, 45% and 25%, respectively), which can substitute cereals and fat, thus providing a meat-like structure [21,22]. This protein source was first discovered in 1966 and there is evidence from human studies that this protein can contribute to satiety, improved metabolic pathways, maintain healthy blood cholesterol levels, and contribute to a better muscular protein synthetic response [21,23]. However, some controversy seems to be arising, as recent studies reported that this food product can be responsible for allergic and gastrointestinal reactions [24]. 

Microbial protein from bacteria is also considered a very promising protein source, with contents ranging from 50–83% of the dry biomass, where production can be attained through renewable energy and direct air capture of CO2 using H2-oxidizing bacteria [25]. However, there are several challenges to overcome, mainly related to the optimization of cultivation conditions (i.e., avoid contaminating microorganism), assessment of nutritional and safety issues (e.g., nutritional, toxicological and allergy assessment), and the need of downstream processing after cultivation. It is important to understand long-term clinical health effects of consuming a diet containing microbial proteins. Development of specific allergies will be a natural consequence of a widespread consumption [21,26]. In addition, microbial biomass has associated a high nucleic acid content (i.e., RNA), which promotes adverse health effects, thus requiring postprocessing, and subsequently, increased costs for its removal prior to human or animal consumption [16]. Alongside the production, recovery, and processing challenges, the consumer perception is ultimately the most critical factor towards the widespread adoption of microbial protein, considering microorganisms are generally also associated with spoilage and disease. 

### 2.3. Insect Protein

Insects as food is a very recent concept in western societies, where there is an incorrect perception linking the practice with poverty and filth. Entomophagy (technical term that refers to insects’ consumption) is popular and very ancient in populations across Asia, Oceania, and Africa. Up to 2,000 species are documented—some of the most popular include caterpillars, ants, flies, beetle larvae, crickets, moths, beetles, and grasshoppers, among others—and their protein content can go up to 76% of dry weight with a balanced composition of EEA [27,28]. This protein source brings lesser environmental footprints given the following points: (i) short life cycle and high efficiency in biomass conversion; (ii) less requirements of water; (iii) no competition for arable land; and (iv) lower pollutant emissions [29]. Nonetheless, many efforts still need to be made to surpass several hurdles regarding safety, legal frameworks, and public perception. Some insects may contain toxic bioactive compounds (e.g., pesticides and heavy metals), and like other proteins, induce adverse human allergic reactions [28]. A multiactor approach engaging government policies, industry, academia, and society is needed for marketing and public acceptance [30].

### 2.4. Algae Protein

Algae includes a diverse group of prokaryotic (cyanobacteria) and eukaryotic, unicellular (as case of microalgae) or multicellular (as case of macroalgae), autotrophic, photosynthetic, and aquatic organisms. 

Algal protein biomass encloses several advantages over conventional meat and plant sources [31,32,33], as follows: (i) fast and high production yields with high photosynthetic efficiency; (ii) no competition for arable land and potable water; (iii) ability to neutralize carbon emissions; (iv) flexibility of their metabolic ways—through auto or heterotrophic processes; (v) biodiversity and possibility of a sustainable harvesting. Macroalgae, or seaweeds, were part of human diet in Asian countries for centuries, and may contain up to 50% of protein on a dry weight basis [34]. Species such as *Porphyra tenera* and *Palmaria palmata* present significant protein content (ranging from 13 to 24% on a dry mater basis) and a pleasant flavor, which is also appreciated in western countries [35,36]. However a main limitation regarding the recovery of protein rich-fractions from macroalgae is related to the presence of large quantities of high-viscosity anionic polysaccharides (e.g., alginate and carrageenans), which can make difficult the solubilization of protein fraction during extraction treatments [36,37].

The cyanobacteria *Arthrospira spp.* (such as *Arthrospira platensis* and *Arthrospira maxima*) attracts considerable interest in the food industry due to its high protein content, which can range from 46–63% (dry matter), presenting all necessary EAA and good digestibility profiles; thus, it is an outstanding alternative to proteins from meat and soybean [38,39,40]. Microalgae *Chorella spp.* also presents a considerable amount of protein—around 50% of dry matter—and together with *Arthrospira spp.* is one of the most employed in commercial microalgae-derived foods [41]. 

Although algae are considered a promising source of proteins, there are some limitations to human consumption. The existence of cell walls rich in cellulose makes gastrointestinal digestion unfeasible by restricting the access of digestive enzymes, thus impairing nutritional and functional value of some algae proteins [42,43]. The cellulosic cell wall can represent up to 10% of the algal dry matter, and given the lack of human enzymes to digest it, several physical and chemical treatments are being developed to permeabilize the cell wall, thus allowing to increase bioavailability of these proteins [40]. This implies that compounds of interest must be extracted, or alternatively, these cellular structures should be previously digested or weakened. This in turn, leads to a greater intensification of “downstream” processing, which can represent high costs on a large scale [38]. 

## 3. Downstream Processing

Several systematic reviews focused on the use of novel and emergent downstream processing technologies applied to the extraction of functional compounds from algae biomass and food waste streams. These technologies should be affordable, guarantee efficiency, and contribute to the quality and safety of the obtained products, but also bring environmental benefits. Process integration/intensification and the need of reducing both consumption of nonrenewable resources (i.e., water and fossil fuels) and pollutant emissions are bringing an opportunity for the development of so-called green processing strategies. Novel and emergent downstream processing includes electric fields, ultrasounds (US), electromagnetic (MW), high hydrostatic pressure (HHP) and enzymatic hydrolysis. These methods combine physical and biological effects to support thermal treatments (e.g., sterilization), hydrolysis, and permeabilization of cellular tissues (i.e., through electroporation) for enhanced extraction of intracellular components. These technologies claim enhanced efficiency and reduction in the operational costs with benefits in the final quality and safety of the products. However, given their high investment costs, a successful implementation is always dependent on economic feasibility, which should be analyzed on a case-by-case basis.

### 3.1. US

US is recognized by its cavitation effects—creation of bubbles followed by their collapse—due to the delivery of high-frequency sound waves. US action promotes structural changes at cellular level without the need for an excessive thermal load, thus reducing use of organic solvents or pH-shift methodologies. US can reduce the size of protein aggregates—e.g., pea protein isolate, soy protein isolate—due to cavitation effect and its hydrodynamic shear forces, resulting in different emulsifying properties [44]. This technology was investigated for the extraction of proteins from several sources such as microalgae and insects, using US action alone or in combination with other strategies such as solvent phase partitioning [45,46,47]. 

### 3.2. MW

MW uses electromagnetic radiation with frequencies ranging from 300 MHz to 300 GHz, resulting in dielectric heating in due to energy generated by dipolar rotation and ionic conduction [48]. Electromagnetic waves interact within different macromolecules at different levels depending on their dielectric properties, thus determining the amount of energy that is reflected, transmitted, and adsorbed. Microwave electromagnetic waves can interact with the folding process of globular proteins, affecting their denaturation pathway, thus holding potential for several biotechnological applications regarding protein functionalization (e.g., protein aggregation) [49]. Given its versatility and ability to produce fast heating, MW was also used in protein extraction processes from different vegetable and biological matrices [50,51].

### 3.3. HHP

HHP consists of applying isostatic pressure, typically between 400–600 MPa, independent of the size and geometry of the product. At these pressures, protein noncovalent bonds can be disrupted (e.g., ionic, hydrophobic, and hydrogen bridges), thus affecting mostly quaternary and tertiary protein structures [52,53]. Despite being considered a nonthermal technology, temperature increase can occur due to adiabatic compression. The combination of these effects (pressure and heat) can also result in protein fractions with different functional and technological features. HHP is viewed as an important tool on protein functionalization by contributing to changes on mechanisms of protein unfolding and aggregation, resulting in different gelation, foaming capacity, and emulsifying properties [54]. For instance, HHP treatments in peas (at 600 MPa and 60 C) can increase protein digestibility up to 4.3% [55]. More recently, it was reported that HHP can also increase enzymatic activities resulting in higher degree of hydrolysis of protein isolates from flaxseed and kidney bean [56,57].

### 3.4. Electric Field Processing—A Promising Approach

Electric field-based technologies find several high-potential applications and are considered to be one of the most important food processing technologies of the future [58,59]. These technologies use electricity and depending on the way how electric wave is delivered, heat can be generated within food products (e.g., ohmic heating) and/or also induce nonthermal events (e.g., electroporation). A proper combination of thermal and electrical events can assure disruption cellular structures without damage integrity of target fractions. This versatility offers a wide range of biotechnological applications regarding protein functionalization when compared with that of other emergent or novel technologies.

Ohmic heating (OH), Pulsed Electric Fields (PEF) showed promising results in downstream processing of food waste and micro- and macroalgae, discarding or reducing the need of using mechanical (e.g., bead milling) or chemical extraction. These technologies are in line with governmental polices for the reduction of environmental footprint, claiming the following reasons [29,60]: delivered electricity can be generated by a renewable energy source (e.g., photovoltaic or hydroelectric power); heat is volumetrically generated inside the product through an efficient way, where in general 98% electricity is converted into heat; there is no need to transport and transfer heat by conduction or convection mechanisms, thus eliminating the use boilers and steam generation systems; by discarding the use of boilers and steam is possible to reduce water consumption and wastewaters; PEF technology by replacing heat production discards the use of cooling systems, thus representing energy saving and less pollutant emissions.

In particular, electric-field based technologies, such as OH and PEF showed promising results with extraction of compounds, including proteins from microalgae biomass [61,62,63,64]. More recently, novel perspectives about the influence of electric fields on protein structure were unveiled. Electrical variables (such as electric field intensity and frequency) can be combined with temperature, and boundary conditions (pH, ionic strength, and protein concentration) to tailor structures at nano- and microscale, affecting macroscale properties of protein-structured systems, such as rheological behavior of gels and emulsions as well as permeability properties of protein-based films [65,66,67]. β-lactoglobulin (β-lg)-rich fractions such as whey protein isolate were used as model and evidenced that those conformational disturbances during electric field application can give rise to gels with different rheological properties [68,69]. OH was also used to produce milk acids gels with different texture and water holding retention [70,71]. OH effects were also reported to change functional properties of soybean protein fractions such as: free amino acid content (increased up to 14%); solubility (increased up to 10%); foaming activity (reduced by 10 to 40%); foaming stability (reduced from 8 to 28%); emulsifying activity (increased up to 38%); and emulsion stability (decreased by 65%) [72]. OH is also demonstrating promising prospects regarding potential alterations on the allergenicity of some protein fractions [73,74]. Immunoreactivity of protein fractions from soybean and β-lg to their specific antibodies can be changed depending on heating kinetics as well as electrical frequency applied [75,76].

OH offers also an opportunity to interact with protein structure through three different ways: (i) heating kinetics (through joule effect); (ii) electrical events (e.g., electrostatic and polarity disturbances); and (iii) occurrence of electrochemical events (e.g., electrolysis). Through these different pathways it is possible to tune protein unfolding and denaturation aiming intended applications [77,78]. One promising field of application of electric field effects is the tailoring the production of protein nano- and microstructures in a way that encapsulates bioactive compounds for an intended functional delivery.

## 4. Structured Systems for Functional Food and Health

Food protein-rich fractions (either animal or nonanimal) present added value given their functional properties such as gelation, foaming, and emulsifying capacities. These properties can be properly tuned to develop biomolecules carrier systems (at nano- and microscale) with intended nutritional and bioactive functionalities at human body. Milk proteins, given their well-known functional and technological properties, were used for decades in development of innovative solutions in food, cosmetics, and pharmaceutical areas. The use of dairy products by reducing the need of expensive downstream processing—such as purification and dehydration operations—is considered a promising strategy to reduce production costs without impairing functional value [29]. Another promising protein source is the one from algae, which seems to gather high potential for development of functional food systems and biomaterials. Microalgae biomass encompasses a great biodiversity and richness in its bio-composition, but it is still unexplored. Available reviews are much more concerned with macronutrients production yields or downstream processing technologies, aiming at fractionation rather than characterizing obtained fractions [79,80]. Proteins resulting from plant-based products—such as the ones from pulses, cereals, and oilseeds—are now in the spotlight of food arena given the need of sustainability. However, studies regarding their functional properties are still scarce. This can be explained by several reasons, such as relatively low protein contents, and low yields of recovery and purity. Despite the great interest of other emerging proteins sources such as the ones from macroalgae and other microorganisms (e.g., fungi and bacteria), the ones from milk byproducts, and microalgae are expanding their applications, thereby showing themselves to be promising alternatives in the development of innovative systems in food and biomaterial science. Table 1 and Table 2 briefly summarize some of the most recent studies on the technological potential of milk protein fractions and microalgae, respectively. In the following subsections the most recent prospects about functionality of protein fractions and potential to assemble protein networks are overviewed, given particular emphasis to the ones milk byproducts and microalgae.

### 4.1. Milk-Derived Proteins

Nondefatted (ND) liquid whey protein products can be used as raw material to produced acid gels either by advanced processing, fermentation (using lactic acid bacteria) or by glucono-δ-lactone (GDL) acidification, presenting an interesting viscoelastic behavior, functional, and nutritional properties Whey protein nanofibrils with a high surface hydrophobicity can be produced and used as nanocarriers to improve the aqueous solubility of curcumin at pH 3.2 [84]. Biodegradable WPI based-biocomposites were used as pH-sensitive controlled delivery systems of antimalarials, such as proguanil hydrochloride and chloroquine diphosphate; the presence of WPI allowed to enhance the swelling capacity of the produced hydrogels, contributing to the retention of a substantial amount of drug in simulated gastric juice condition, and thus, offering a promising potential for controlled delivery of these bioactive agents [85]. Liquid whey concentrates were used as primary material to produce GDL gels which presented a stronger network when compared with that of ones resulting from fermentation, and that cold storage, as well as the use of the skimmed milk proteins as additive, could help to improve rheological properties [86]. Liquid whey can be used as basis to produce other food products such as whey cheeses with added kefir or probiotics [87]. In fact, fermentation-based solutions are an emergent hot topic for the reuse of dairy wastes and production of bioactive proteins and peptide fragments. These bioactive compounds, such as lactoferrin and bioactive peptides, can be incorporated into the formulation of structured systems, such as edible films, and thus, contribute to active packaging by enhancing preservation of perishable food products [88]. 

There is also growing interest on the use of milk proteins as building blocks for the development of encapsulation systems and scaffolds for tissue engineering. Nascimento and coworkers [89] recently reviewed the use of casein and casein-based solutions for the development of hydrogels for biomedical applications. This review includes the description of several type of dispersions (such as emulsions and suspensions) and gels, emphasizing their use for the entrapment of molecules such as vitamin B5, Nile red, or ibuprofen [82,83]. Farooq and coworkers [90] summarized the use of whey proteins for the development of nanoemulsions, nanosuspensions, hydrogels, and microparticles using WPI and WPC as excipients for the delivery of several molecules. 

Agnieray et al. [91] recently reviewed the use of sustainably sourced protein-based biomaterials, highlighting milk proteins from whey and casein fractions, among others. They prospected that such proteins offer myriad of advantages that included versatility and flexibility of application, tunability, biodegradability, as well as increased biocompatibility and cytocompatibility. These properties make protein-based materials excellent biomaterials for scaffolds production aiming tissue engineering applications [92]. Scaffolds for tissue engineering can be produced using BSA (4.19%, w/v) and casein (0.69%, w/v) through the Ca^2+^-induced “cold gelation” procedure [93]. A recent study unveiled that a proper optimization of the heating conditions during the scaffolds’ production can avoid the use of reducing agents such as ditiotreitol (DTT) [4]. Figure 2 evidence microstructure of BSA/casein scaffold produced under conventional heating and ohmic heating, highlighting the importance of the heating process and presence of an electric field on the obtainment of singular protein networks. These “clean label” scaffolds promoted improved cellular growth and presented no cytotoxicity to BJ-5ta fibroblast cells. Khanna and coworkers [94] produced BSA thermo-responsive hydrogels upon heating high concentrations of BSA protein (14 to 22% w/v) at 65 °C for less than 3 min. This procedure allowed to obtain fibrillar structures resembling amyloids aggregates cytocompatible with HaCaT skin cell lines, and thus with potential applications for topical drug delivery applications.

### 4.2. Microalgae Proteins

Microalgae protein biomass includes interesting bioactive compounds with interest of application in several areas, such as food and pharmaceutical industry [95,96,97]. However, it is important to address technological functionality of these fractions and how they can be used. Downstream processing, which involves operations such as extraction, fractionation, purification, and concentration, is often necessary to obtain a functional concentrated protein extract that can comprise both nutritional requirements and specific technological functions [80].

Microalgae suspensions may present different rheological properties depending on the microalgae strains and its cell properties, such as cell wall composition and structure [98]; in this respect, downstream processing such as US can disrupt cells’ clusters or promoting release of intracellular components changing the rheological properties of microalgae suspensions [99]. Bertsch et. al. [97] recently provided a systematic review on the role of crude protein fractions on the stabilization of fluid interfaces, which shows that several strains of microalgae and cyanobacteria (e.g., *Arthrospira spp* and *Chlorella spp*) can offer protein extracts with ability to adsorb at fluid interfaces (foaming and emulsion formation) that are very comparable to common food stabilizers such as WPI, soy flour, and caseinate. Through an optimized, controlled bead-milling of *Tetraselmis suecica,* it was possible to obtain a soluble functional protein fraction with a surface activity (foaming and emulsification) and gelation behavior superior to that of whey protein isolate. This better performance is supported by presence of glycoproteins and charged carbohydrates as well as lipids in the crude extract [33].

Grossmann et al. [100] used protein extract from the microalga *Chlorella sorokiniana* to produced heat-induced gels. At concentration of 9.9 g/100 mL and a temperature of 61 °C it was possible to obtain a stable gel network, which can be impaired by the increase of the ionic strength of the system. Authors concluded that purification of the extract by reducing the mineral content may strengthen the protein gel network.

Despite its remarkable protein content, *Arthrospira platensis*, known as Spirulina, is still unexplored regarding its potential functional properties regarding gelling and emulsifying properties of their protein fractions. Protein isolate from *A. platensis* can be used to develop elastic heat-induced gels during heating to 90 °C, using a concentration 1.5 and 2.5 wt % in 0.1 M Tris buffer, pH 7, with 0.02 mol/L CaCl2 [101]. These latter authors observed that the network elastic moduli and elasticity can be further strengthened after cooling step, and that exposing hydrophobic regions during heat treatment is critical to trigger gelation process. More recently, it was observed that *A. platensis* protein isolate can be used to produce gel structures in distilled water at 12% (*w/w*), and that the production of structured systems are favored at pH 10 [102]. Another study showed that, independent of the level of protein purification, the extracts from *A. platensis* can help stabilize emulsions; however, authors also conclude that purification can improve functionality by yielding smaller emulsion droplets and stronger viscoelastic networks at the oil–water interface [103]. Contrary to what happens to common animal-based proteins such as milk, proteins from microalgae biomass can act as emulsions and foams stabilizers in a broad range of pH and ionic strength due to their characteristically low isoelectric point (3–4) [97]. 

Purification of microalgae protein fraction also allows to recover important compounds that can be used in the development of biomaterials. For example, phycobiliproteins and glycoproteins present interesting bioactivities—i.e., antitumor, antimicrobial, anti-inflammatory, UV protection—finding several commercial applications not only in food and feed, but also in nutraceuticals, medicines, cosmetics, and dyes [104]. Regarding macroalgae, the great interest in its biochemical composition is related to their polysaccharides fraction [105].

Another recent prospect is related to the potential of using microalgae biomass as whole-to-structure foods. From industrial point view, this would be an attractive and sustainable approach once extensive downstream operations can be avoided (reducing waste streams such as solvents or biomass); furthermore, the bulk material includes structural polysaccharides and bioactive components that can contribute to harnessing food structuring. This would allow us to enhance nutritional and health effects, thus giving rise to a multifunctional ingredient [79,106]. Bernaerts and coworkers [106] studied the rheological behavior of several commercially microalgae species in aqueous suspensions. They concluded microalgae suspensions of *Arthrospira platensis* Schizochytrium sp. and *Phaeodactylum tricornutum,* show shear-thinning flow behavior at the concentration of 8% *w/w*, whereas *Chlorella vulgaris, Porphyridium cruentum*, and *Odontella aurita* suspensions resulted in development of weak gels, showing most significant properties as structuring agents. Latter authors also concluded that rheological properties of suspensions can be largely influenced by pH and processing such as thermal treatments and high-pressure homogenization, resulting in increased storage modulus and viscosity in some microalgae suspensions (e.g., *Chlorella vulgaris and Arthrospira platensis*), whereas the opposite behavior was observed for *Porphyridium cruentum* and *Odontella aurita*. 

### 4.3. Emergent Sources

Several other protein sources from nonconventional or underexplored sources demonstrated potential on de development of functional and structured food systems. Underused animal proteins can be recovered from waste and used as technological agents. Proteins recovered from the wastewaters of food processing plants, including fish and meat slaughterhouses and transformation plants, displayed potential to be used as emulsification and water holding agents in food technology applications [110]. The use of mannoprotein recovered from brewer’s yeast demonstrated good emulsifier/stabilizer capacity and even scored high in terms of color, taste, flavor, and overall acceptance when applied on a salad dressing [111].

The use of plant proteins is now a reality in the food industry and a growing number of successful implementations was reported either in new applications or as a replacement for conventional protein systems. Furthermore, the use of innovative processing strategies to enhance these emerging protein sources functionality was considerably reviewed [112]. These strategies decisively boost emerging protein techno-functional properties. For instance, oilseed residue, resultant form oil productions, was identified as a promising protein source. Although the high-protein content and successful recovery strategies were reported, the functionality of these proteins is often limited. Processing sunflower protein with EF, the protein´s secondary and tertiary structures and thermal properties were changed. These changes were reflected on techno-functional properties such as particle size and surface properties of the protein ingredient [113]. Other techniques, such as enzymatic hydrolysis, demonstrated potential in improving the techno-functional potential of emerging protein sources. Using this approach in protein recovered from defatted pigeon pea milling promoted its gelling ability, which was not observed in the nontreated isolate [114]. 

The development of innovative protein-based structured systems from emerging sources is still on its infancy. However, with the growing repertoire of potential protein sources and processing technologies available, it is expected in the near future that there will be an exponential growth in developed solutions and implementation in the food industry. 

## 5. Conclusions and Future Perspectives

The body of knowledge about the characterization and the use of sustainably sourced proteins as functional biomaterials in food and biomedical applications was expanded in recent years. Protein-based materials from dairy byproducts and microalgae biomass gather promising prospects of application. They can be considered sustainable materials given their chance of valorization and integration on a biorefinery context. They also offer a wide range a bulk of bioactive molecules and an interesting nutritional composition that can be used for nutritional supplementation. Nonetheless, these products are only profitable if the number of unit operations for their obtention is reduced, and this should be attained without compromising intended functionality. In this context, downstream processing should be used as strictly as necessary, and/or elegantly designed, thus envisaging process integration through biorefinery strategy as way to reduce costs. Extraction, separation, fractionation, and purification steps often impair an economy of scale and/or affect negatively technological and functionality of the final products, and because of that, should be efficiently used. Electric-field based technologies comprise important advantages at both environmental and processing levels. The possibility of applying several physical and chemical effects (i.e., heat, electricity and electrochemical) through an independent or combined way increases versatility and the possibility of process integration. Moreover, processes such as PEF and OH are already available at commercial scale in food processing, which make their use much more straightforward for other applications. 

Other bottlenecks are linked with scaling-up and lack of systematic information regarding functional and health claims, which include safety assessment (e.g., allergenicity and toxicity) and bioaccessibility issues. Many of the microbial protein fractions are still poorly characterized, lacking information about their structural aspects and interactions with other molecules, unlike what happens with structured systems based on common food proteins, such as those from milk. Generally, it seems evident that functional properties of algae protein (e.g., gelation) result from protein complexes. It is crucial to characterize these complexes and understand the role of polysaccharides in interfacial properties of proteins. Macroalgal polysaccharides (e.g., alginate, agar, and carrageenan) are between the most important compounds incorporated in food products. A better fundamental understanding about protein–polysaccharide interactions may increase the chance of developing biodegradable structured systems with tunable properties for the improvement of the food texture. Another recent trend focused on fermentation-based solutions. The use of beneficial microbial consortia is likely to become a common pathway to produce protein-based biomaterials driven by health benefits and sustainability issues. Lab-grown food, commonly known as cellular agriculture, is now in its onset of expansion as a promising way to sustainably provide alternative food products and nutrients. The exploration of plant-based sources and algae to produce functional proteins via fermentation, as well as valorization of underused conventional protein fractions, such as the ones from dairy byproducts, will gain traction in forthcoming years.

## Figures and Tables

**Figure 1 gels-07-00161-f001:**
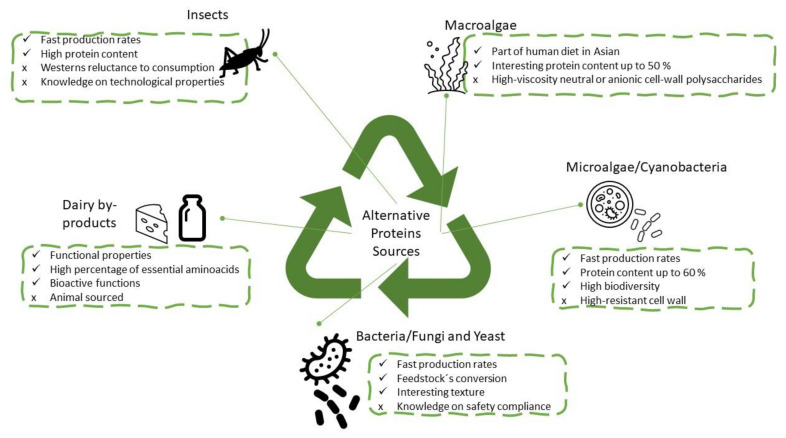
Examples of alternative and sustainable protein sources and some of their main features.

**Figure 2 gels-07-00161-f002:**
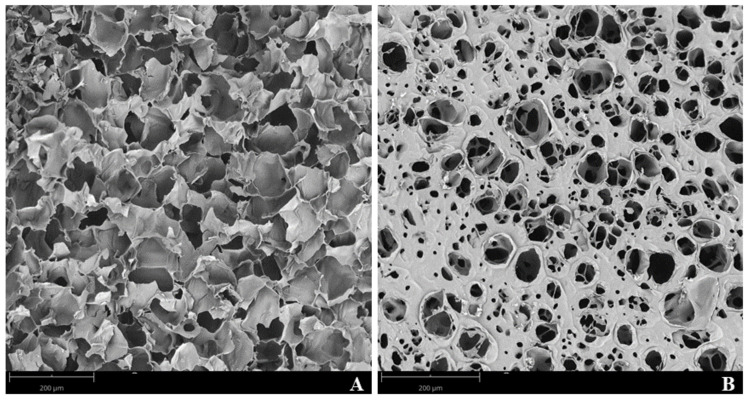
Scanning electron microscopy (SEM) micrographs of the scaffolds surface produced under conventional heating (**A**) and ohmic heating (**B**) [4]. Scale bar corresponds to a size of 200 µm.

**Table 1 gels-07-00161-t001:** Recent studies about technological potential of milk proteins.

Source	Functional Properties	Application	Reference
Milk protein concentrate and whey protein concentrate	3D printing of food simulant	Food	[81]
Casein nanocarrier	Glyceraldehyde (GAL) used as a crosslinking agent for the entrapment of Nile Red	Biomedical	[82]
Casein-based nanocomposite	Favorable for drug loading and release using ibuprofen (IBU), docetaxel (Dtxl), and vitamin B5 (B5) as model	Biomedical	[83]
WPI nanofibrils	Complexation with curcumin	Food	[84]
WPI-based nanocomposites	Controlled delivery of antimalarials	Biomedical	[85]
BSA/casein	Production of biodegradable scaffolds	Biomedical	[4]

**Table 2 gels-07-00161-t002:** Recent studies about technological potential of protein form microalgae.

Source	Functional Properties	Application	Reference
*Tetraselmis suecica*	Improved gelation and surface activity (foaming and emulsification) when compared with WPI	Food	[33]
*Chlorella sorokiniana*	Protein heat-induced gel governed by electrostatic interactions	Food	[100]
*Chlorella sorokiniana and Phaeodactylum tricornutum*	Emulsifying properties of water-soluble proteins	Food	[107]
*Spirulina sp. and Isochrysis galbana*	Water- and oil-holding capacities, foaming, emulsifying activities, and stabilities of protein extracts	Food	[108]
*Arthrospira platensis*	Protein isolate with emulsifying, foaming, gelling, and film-forming properties favored at pH 10	Food	[102]
*Arthrospira platensis*	Interfacial viscoelastic network was faster, and final network strength increased with the degree of purification	Food	[103]
*Chlorella vulgaris*	Enhanced emulsifying capacity and stability	Food	[109]
*Porphyridium cruentum, Chlorella vulgaris and Odontella aurita*	Shear thinning behavior at 8% *w/w*, showing aelastic-like behavior (*G′* > *G″*)	Food	[106]

## Data Availability

Data sharing not applicable.

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
