# Peer review of "Emergent Proteins-Based Structures—Prospects towards Sustainable Nutrition and Functionality"

_gels, 2021, doi:10.3390/gels7040161_

Round 1

Reviewer 1 Report

The manuscript entitled ‘’Emergent proteins-based structures – prospects towards sustainable nutrition and functionality’’, deals with novel protein sources for food and biomaterials. The topic is of high interest at the moment for researchers and industry. The authors have nicely structured the review and critically discussed available literature. The number of publications reviewed seemed appropriate. However I have some comments that the authors should address before the manuscript could be consider for publication.

-There are existing reviews in this area , for example: Annual Review of Food Science and Technology Volume 10, 2019 , pp 311-339 by Loveday , and Current Opinion in Food Science Volume 32, 2020, Pages 156-162. Food proteins from plants and fungi. Both reviews deal with emerging proteins, the authors should explain how theirs complements those ones. How this review contributes to other existing reviews in the field?

-Line 26_  How is Covid pandemic accelerating food demand and in particular for alternative proteins? The authors should be more specific or remove the sentence.

-Line 67_ Plant proteins can also lead to allergies_ the sentence is speculative. Regarding low cost , how is this compared to animal proteins ? Any references?

-Minor language/typos need to be checked in several places throughout the manuscript.

-Line 81_ Macroalgae are not microorganism.

-Line 162_ How does the cellulose content compare to fruits and vegetables and legumes? The cell wall is a limiting ‘barrier’ for all plant materials.

-Figure 1: There are some typos in the text. Maybe would be clearer to mark the features as pros/cons?

-Line 176- Do these technologies have any constrains or negative implications?  

-Line 354-The section is about microalgae, this sentence seems out of focus.

-Line 356- This reviewer has done a quick search and found more recent studies on microalgae and macroalgae as whole ingredient for food applications as function of concentration and other processing methods, see below, the authors should consider including more recent publications:

Bioresour Technoly . 2013 ;139:209-13.Influence of cell properties on rheological characterization of microalgae suspensions. Zhang  et al.

Algal Research Volume 49, August 2020, 101960 Composition and rheological properties of microalgae suspensions- Martinez Sanz et al.

Food Hydrocolloids 120, November 2021, 106989 Food Hydrocolloids. Macroalgae suspensions prepared by physical treatments: Effect of polysaccharide composition and microstructure on the rheological properties. Malafronte et al.

-Line 416. The topic of lab grown food has not been discussed in the main body of the review, it should not be in the conclusions in this review.

-Line 417 -Fermentation should be discussed to a larger extend if it is included in conclusions: what is known about fermentation of micro and macroalgae?

Reviewer 2 Report

The manuscript ‘Emergent proteins-based structures – prospects towards sustainable nutrition and functionality’ is an overview of recent research in technological functionality of underused protein fractions, recovered from microbial biomass and food waste sources. The paper is an interesting review, but some improvements should be applied before publication in Gels.

Minor comments:

l. 55 – should be ‘nonconventional sources of proteins’

Major comments:

The text of the manuscript should be presented as a complete review through recent research with a logical structure: motivation, sources, technologies, application of technologies to available source.

The title doesn’t correspond to the text of the manuscript. Authors limited analysis to food waste, microbial protein and algae protein. Title gave an impression that the subject of the paper is wider than in reality

As a sources, authors mentioned three categories: food waste, microbial protein and algae protein, but food waste is removed from discussion presented in section ‘Structured systems for functional food and health’ Please add missing discussion

Technological approach (section3) correspond only to one technology – electrical approach. Authors should extend analysis other technologies than electric field based, and improve the discussion in section 4.

Round 2

Reviewer 1 Report

The authors have done an excellent job addressing my comments. My recommendation is Accept for publication.

Reviewer 2 Report

Dear Autors,

the manuscript is suitable for publication inpresent form.